# Lipidomic and metabolomic profiles of *Coffea canephora* L. beans cultivated in Southwestern Nigeria

Chinyere F. Anagbogu[1,2,3]*, Jiaqi Zhou[1], Festus O. Olasupo[2,3], Mohammed Baba Nitsa[3], Diane M. Beckles[1]*

**1** Department of Plant Sciences, University of California, Davis, CA, United States of America, **2** Department of Crop Protection and Environmental Biology, University of Ibadan, Ibadan, Nigeria, **3** Crop Improvement Division, Cocoa Research Institute of Nigeria, Ibadan, Nigeria

* flora2na@yahoo.com (CFA); dmbeckles@ucdavis.edu (DMB)

**Data Availability Statement:** Data underlying the results presented in the study are available from https://doi.org/10.5061/dryad.j6q573nc4

## Abstract

Coffee (*Coffea* spp.) is one of the most popular refreshing beverages globally. Coffee lipid diversity has untapped potential for improving coffee marketability because lipids contribute significantly to both the health benefits and cup quality of coffee. However, in spite of its potential importance, there have not been extensive studies of lipids among *C. canephora* genotypes. In this study, ultra-performance liquid chromatography coupled with mass spectrometry (UPLC–MS) profiling of lipid molecules was performed for 30 genotypes consisting of 15 cultivated and 15 conserved genotypes of *C. canephora* in Southwestern Nigeria. We identified nine classes of lipids in the 30 genotypes which belong to the 'Niaouli', 'Kouillou' and 'Java Robusta' group: among these, the most abundant lipid class was the triacylglycerols, followed by the fatty acyls group. Although 'Niaouli' diverged from the 'Kouillou' and 'Java Robusta' genotypes when their lipid profiles were compared, there was greater similarity in their lipid composition by multivariate analysis, compared to that observed when their primary metabolites and especially their secondary metabolite profiles were examined. However, distinctions could be made among genotypes. Members of the fatty acyls group had the greatest power to discriminate among genotypes, however, lipids that were low in abundance e.g. a cholesterol ester (20:3), and phosphotidylethanolamine (34:0) were also helpful to understand the relationships among *C. canephora* genotypes. The divergent lipid profiles identified among the *C. canephora* genotypes, correlated with their Single Nucleotide Polymorphism grouping as assessed by genotype-by-sequencing, and will be exploited to improve coffee cup quality.

## Introduction

"Lipids" are any organic compounds that are insoluble in water, but soluble in organic solvents [1]. Represented in this group are a broad range of molecules such as fatty acids, triacylglycerols, and phospholipids. Lipids serve diverse and critical biological roles in plants such as

**Funding:** Funding was provided in part by the Norman E. Borlaug Leadership Enhancement in Agriculture Program (Borlaug LEAP) through a grant to the University of California-Davis by the United States Agency for International Development and, the Ministry of Agriculture in Nigeria through a grant to the Cocoa Research Institute of Nigeria. The funders had no role in the study design, data collection or analysis. The funders had no role in decision to publish, or in the preparation of the manuscript.

**Competing interests:** The authors have declared that no competing interests exist.

maintaining cellular membrane integrity and homeostasis, acting as key components in cell signaling pathways and being used as an energy store [2]. In coffee beans, lipids are important reserve compounds and can make up to 10–17% of fresh weight [2, 3].

The lipids accumulated in coffee beans are important contributors to beverage organoleptic properties [4–6], and are excellent discriminators of coffee quality [7–11]. Lipids determine beverage foam and emulsion formation and stabilization, which in turn influences flavour and aroma enhancement, especially in espresso brewing [4, 12]. Coffee foam and emulsions are responsible for the characteristic pleasant aroma of coffee by trapping and retaining the volatile compounds [13]. Further, the foam created during coffee brewing also influences beverage creaminess and texture [14, 15], which further enhances coffee sensory perception.

Coffee bean lipid composition is influenced by the environment e.g. altitude, shade and temperature [8, 9, 14, 16] but is largely genotype-dependent [11, 17]. Differences are apparent in *C. arabica* and *C. canephora*, the two most consumed types of coffee [11]. Arabica coffees generally have lipid contents of 15% which is higher than that in Robusta which averages 10% [2, 3]. Arabica also has higher palmitic (16:0), arachidic (20:0), stearic (18:0) and linolenic acids (18:3), but is lower in oleic acid (18:3) compared to *C. canephora* [18, 19].

The value of drawing correlations between the metabolomic and genetic profiles of plant tissues have been comprehensively discussed by Harrigan, *et al.* [20] and Beckles and Roessner [21]. Knowledge of the metabolome can help to cement the gap between genotype and phenotype [22]. Identifying genetic polymorphisms in the genes encoding components of metabolic pathways, is a new area of interest to scientists, including researchers working to improve cup quality in coffee [6]. A recent genome wide association study of 107 *C. arabica* accessions found single nucleotide polymorphisms positioned within or near genomic regions coding for proteins involved in lipid and diterpenes metabolic pathways [23], indicating the potential genetic diversity of these key metabolites.

Lipidomics in plants is still in its infancy compared to the broader metabolomic profiling, and the data for *C. canephora* is comparatively less than that available for *C. arabica*. *C. arabica* is known to have a high lipid content compared to *C. canephora* [2, 3]. However, there have not been extensive studies of lipids of *C. canephora* genotypes, so the variability relative to *C. arabica* still remains largely undiscovered. This is important because although *C. canephora* is not as popular as *C. arabica*, its production and consumption is increasing, making it important to understand its organoleptic properties [24]. Further, genetic diversity among tetraploid *C. arabica* is not as high as diploid *C. canephora* [25] and this may be reflected in the metabolomic and lipid species diversity within these species.

The aim of this study was to determine the lipid and metabolomics profiles of *C. canephora* genotypes conserved in the Cocoa Research Institute of Nigeria (CRIN) coffee repository, and those used as a source of subsistence by farmers in the southwestern region of Nigeria. Because of the multifaceted role lipids play in determining coffee sensory quality, we wished to identify potential lipid markers differentiating among these *C. canephora* genotypes. These markers would be critical for future coffee improvement programs in Nigeria. We also wished to broaden the current understanding of lipid profiles in diverse coffee species for which there is relatively scarce data. An ultraperformance liquid chromatography method that incorporated a charged surface hybrid (CSH) solid phase was used to separate the different lipid molecules, followed by electrospray ionization (ESI) quadrupole time of flight (QTOF) tandem mass spectrometry (MS/MS) i.e. CSH-ESI QTOF MS/MS. These compounds were subjected to univariate and multivariate analyses to understand their relative abundance. They were also compared to the primary and secondary metabolites profiled in these same coffees, and their relatedness based on lipidomics was compared to that based on Genotyping-by-Sequencing data. We therefore were able to build a comprehensive, multidimensional overview of bean lipids in Southwestern Nigerian coffees.

## Materials and methods

### Plant material

Coffee beans (*Coffea canephora)* were harvested from coffee germplasm repository of Cocoa Research Institute of Nigeria (CRIN) and coffee farmers' field and no specific permission was required. Similar genotypes were used for metabolomics [26] consisting of 'Niaouli', 'Kouillou' and 'Java Robusta' varieties. They were classified into six groups based on the result from Single Nucleotide Polymorphism-Genotype by Sequencing analysis [26]. 'Niaouli' is comprised of three genotypes from farmers' field: Nia_1, Nia_2 and Nia_3, (respectively, classified as Groups 1, 2 and 3), 'Kouillou' is comprised of two genotypes from CRIN: C111 and C36, (respectively, classified as Groups 4, and 5) and, 'Java Robusta' is comprised of one genotype from CRIN: T1049, and was classified as Group 6. There were five replicates for each group (genotype) giving a total of 30 samples (Table 1). Both the farmers' fields and the CRIN plantations are in the Southwestern part of Nigeria [26]. They are located at altitudes of 433–522 m and 133 m above sea level respectively, as recorded by ArcGIS software (Redlands, California). Their specific location i.e. longitude and latitude coordinates, is available at Anagbogu et al. [26].

### Sample preparation

Reddish mature (ripened), coffee beans of these genotypes were collected in ice bags and immediately transferred to -80°C. The endosperms of the coffee bean were excised using sterile blade and re-transferred to -80°C. These endosperms were lyophilized, ground into powder with Udy mill (Udy Corporation) and sealed prior to lipidomic analysis. The lipid was extracted following the protocols according to Matyash *et al.*, [27]. Dried extracts containing an internal standard [12-[(cyclohexylamino)carbonyl] amino]-dodecanoic acid (CUDA)] used as a quality control were resuspended with a mixture of methanol/toluene (9:1, *v/v*) (60 μL) [28, 29].

### Data acquisition

All data processing was done at the West Coast Metabolomic Center, University of California, Davis. Extracted lipids were separated on an Acquity UPLC CSH C18 column ($100 \times 2.1$ mm; 1.7 μm) maintained at 65°C. The mobile phases for positive mode consisted of 60:40 ACN: $H_2O$ with 10 mM ammonium formate and 0.1% (v/v) formic acid (A) and 90:10 IPA:ACN with 10 mM ammonium formate and 0.1% (v/v) formic acid (B). For negative mode, the mobile phase modifier was 10 mM ammonium acetate instead. The gradient was as follows: 0 min 85% (A); 0–2 min 70% (A); 2–2.5 min 52% (A); 2.5–11 min 18% (A); 11–11.5 min 1% (A); 11.5–12 min 1% (A); 12–12.1 min 85% (A); and 12.1–15 min 85% (A). Sample temperature was maintained at 4°C in the autosampler. Two μL of sample was injected. Vanquish UHPLC system (ThermoFisher Scientific) was used. Thermo Q-Exactive HF Orbitrap MS instrument was operated in both positive and negative ESI modes respectively with the following

**Table 1. Coffee genotypes used in this study and their symbols.**

| Variety | 'Niaouli' | | | 'Kouillou' | | 'Java Robusta' |
|---|---|---|---|---|---|---|
| Group/Genotype | Group1 (Nia_1) | Group 2 (Nia_2) | Group 3 (Nia_3) | Group 4 (C111) | Group 5 (C36) | Group 6 (T1049) |
| Samples' Symbols | Nia_11 | Nia_21 | Nia_31 | C111_1 | C36_1 | T1049_1 |
| | Nia_12 | Nia_22 | Nia_32 | C111_2 | C36_2 | T1049_2 |
| | Nia_13 | Nia_23 | Nia_33 | C111_3 | C36_3 | T1049_3 |
| | Nia_14 | Nia_24 | Nia_34 | C111_4 | C36_4 | T1049_4 |
| | Nia_15 | Nia_25 | Nia_35 | C111_5 | C36_5 | T1049_5 |

parameters: mass range 120–1700 *m/z*; spray voltage 3.6kV (ESI+) and −3kV (ESI−), sheath gas (nitrogen) flow rate 60 units; auxiliary gas (nitrogen) flow rate 25 units, capillary temperature 320˚C, full scan MS1 mass resolving power 60,000, data-dependent MS/MS (dd-MS/MS) 4 scans per cycle, normalized collision energy at 20%, 30%, and 40%, dd-MS/MS mass resolving power 15,000. Thermo Xcalibur 4.0.27.19 was used for data acquisition and analysis. The instrument was tuned and calibrated according to the manufacturer's recommendations.

## Data processing

Raw data files were converted to the mzML format using the ProteoWizard MSConvert utility. For each m/z values ion chromatogram was extracted with *m/z* thresholds of 0.005 Da and retention time threshold of 0.10 min. Apex of the extracted ion chromatograph was used as peak height value and exported to a .txt file. Peak height files for all the samples were merged together to generate a data matrix. Targeted peak height signal extraction was performed using an R script that is available at https://github.com/barupal. Extracted ion chromatograms for each peak were saved as pictures. CSH-POS and CSH-NEG data matrices were generated. No normalization was applied as minimum signal drift was observed during analysis.

## Statistical analysis

Chroma TOF 4.3X software of LECO Corporation and LECO-Fiehn Rtx5 database were used for raw peaks exacting, the data baselines filtering and calibration of the baseline, peak alignment, deconvolution analysis, peak identification and integration of the peak area [30]. The RI (retention time index) method was used in the peak identification, and the RI tolerance was 5000. Metabolite data were normalized by dividing each peak area value by the area of internal standard (Ribitol). Data were $\log_{10}$ transformed, mean-centered and divided by the standard deviation of each variable before performing statistical analysis. The raw data from the UHPLC has been submitted to Dryad repository and can be found at the following location: https://doi.org/10.5061/dryad.j6q573nc4. The statistical analyses, such as ANOVA, PCA, PLS-DA were performed by using MetaboAnalyst 3.0 [31]. Univariate and multivariate statistical approaches were performed with MetaboAnalystR [31]. One-way Analysis of Variance (ANOVA) test was performed to ascertain the significant variables, and they were expressed as *f*—and *p*-values. The level of statistical significance ($\log_{10}(p)$) was determined, followed by post-hoc analyses to correct the *p*-value and thus generate the False Discovery Rate (FDR). Fisher's least significant difference method (LSD) was used to identify groups that differ in their lipid profiles [31]. The Hierarchical Clustering Analysis (HCA) plots for SNPs and lipids were generated under the R environment. Pearson correlative analysis was performed using lipid and metabolism parameters. Two packages in R were used: "psych" [32] and "reshape2" [33] for calculating Pearson's correlation coefficient (PCC), *p*-value and *q*FDR. The correlation network was drawn using MetScape 3.1.3 [34] in the Cytoscape environment [35].

## Results and discussion

The primary aim of this work was to determine the lipid profile and potential lipid markers differentiating *C. canephora* conserved and cultivated in southwestern Nigeria. The main classes of lipids, their relative abundance among the genotypes and their ability to discriminate among the genotypes were determined. Next, how the genotypes associated with each other based on their primary, secondary and lipid profiles. Finally, the relationship among genotypes based on lipidomics was compared to genotyping-by-sequencing SNP data, to determine if there was good accordance between genotypic and functional data i.e. lipids.

## Identification of lipids

A total of 1824 lipid species consisted of identified (96) and unidentified (1728) lipids were detected with CSH-ESI QTOF MS/MS. The positive and negative ESI operating modes separated the lipids into 1171 positive and 653 negative lipids. Six lipid subclasses are typically recognized in living systems [36], five were found in the *C. canephora* genotypes studied (Fig 1A).

Each class of lipid contains multiple molecules of varying lengths and degree of saturation. Although these lipid classes are diverse, we constructed a simplistic figure of how they may be related to each other in a metabolic network (Fig 1B). We identified a cholesterol ester (20:3) in this work but these compounds have not been well studied in plants.

## Relative amount of lipids variability among genotypes studied

Each lipid group i.e. FA, TG, DG, PC, LPC, is made up of multiple chemicals of varying molecular mass, degree of unsaturation etc. The triacylglycerol (TG) group contained the most

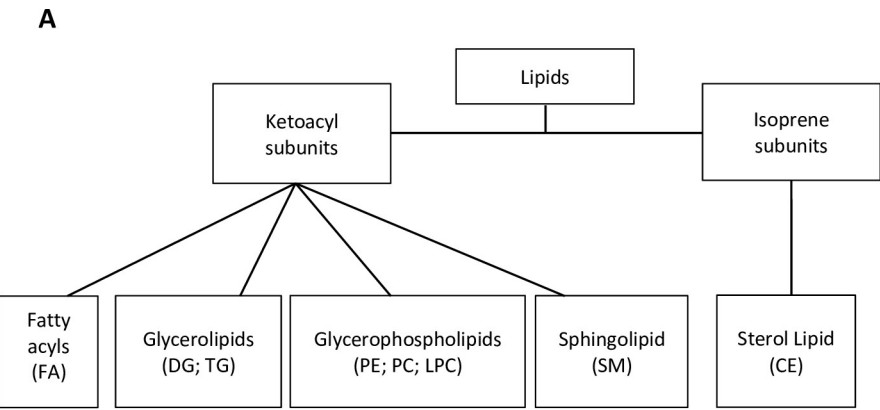

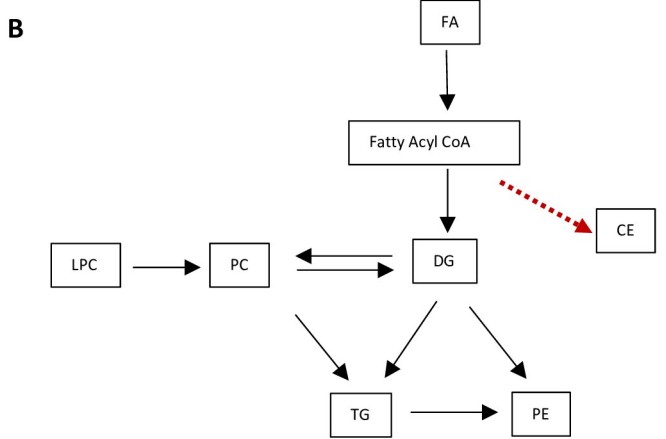

**Fig 1. Lipid classes and their interrelationship. A)** Lipid classes and subclasses detected in *C. canephora* beans. Classification is based on the LIPID MAPS (Metabolites and Pathways Strategy) system, which separates lipids based on their two fundamental "building blocks"—the ketoacyl group and the isoprene group [36]. Abbreviations: FA- Fatty acyls; DG—Diacylglycerol; TG- Triacylglycerols; PE—Phosphatidylethanolamine; PC -Phosphatidylcholine; LPC Lysophosphatidylcholine; SM-Sphingomyelins; CE- Cholesterol esters. **B)** A simplified view of a proposed biochemical interrelationship among the lipid classes according to data in [37, 38], as well as in the MetaCyc database [39] derived from plants. The red dashed arrow indicates data obtained from human cells [40].

chemicals, with 45 different species identified (S1A–S1F Fig). We also identified 13 different fatty acyls (FA), nine phosphatidylcholines (PC), four lysophosphatidylcholine (LPC), three diacylglycerol (DG) species and one each of a cholesterol ether, sphingomyelin and phosphotidylethanolamine.

The relative abundance of the chemicals within each lipid class was examined across genotypes. The ratio of the compounds with the maximum and minimum concentration was determined and was highest for the TG, FA and DG classes i.e. 2145-, 2923- and 20-fold respectively and lowest for PC, LPC, CE, SM, and PE (Fig 2). Triacylglycerol (52:2) and was the most abundant lipid molecule, while Sphingomyelin (SM) was the lowest (Fig 2). the TG and FA were the most abundant lipids in these genotypes consistent with previous reports [2, 3]. Genotype C36_1 ('Kouillou') had the highest amount of TG while Nia_11 ('Niaouli') had the highest amount of FA. The 'Niaouli' (Nia) genotypes are cultivated by farmers, while 'Kouillou' (C111 and C36) and 'Java Robusta' (T1049) are conserved in the CRIN germplasm. To better understand the lipid composition in coffee beans, we examined the specific lipids identified within eight of the nine classes (the unknown compounds were not discussed) and compared them across genotypes. Each class was considered in turn, and all data referenced from 3.2.1 to 3.2.3 is contained in S1 Fig.

**Fatty-acyl group (FA).** The predominant fatty acyls among our *C. canephora* germplasm were linoleic acid (18:2) followed by palmitic acid (16:0) (S1A Fig), which together encompassed 50% of the total fatty acids. Linoleic and palmitic were also the predominant FAs in a range of *C. arabica* and *C. canephora* genotypes studied by Speer and Kölling-Speer [2]. The number of carbons in the fatty acyls in this study ranged from C15 to C28, also similar to the range i.e. C14 to C24, found by Speer and Kölling-Speer [2].

The proportion of the major FA among genotypes was also examined, because of their influence on coffee quality. These FAs are also members of highly related biosynthetic pathways (Fig 3). FA (16:0) i.e. palmitic acid, was higher in 'Kouillou' and 'Java Robusta' compared to 'Niaouli' (S1A Fig). This is of interest because palmitic acid was positively associated with the high hedonistic values in some specialty Brazilian *C. arabica* coffees [9]. Oleic, linoleic, and

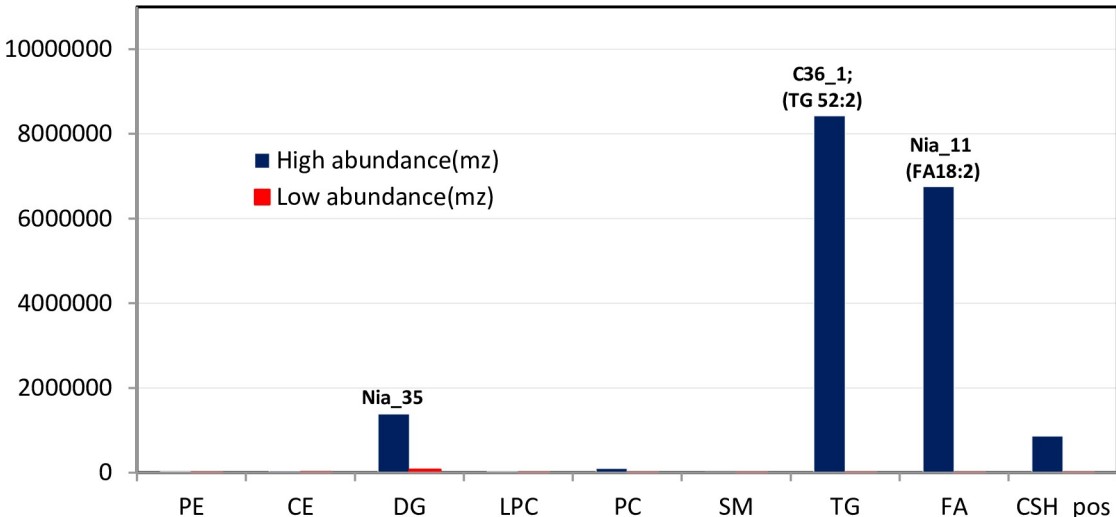

**Fig 2. Relative lipid abundance in the *C. canephora* species studied.** We identified nine subclasses of lipids within the five broader lipid classes described in LIPIDS MAPS. For each subclass, the relative abundance of compounds among each genotype was assessed. Values shown are the maximum and minimum values within each compound class, and the label indicates the genotype in which the highest value was found.

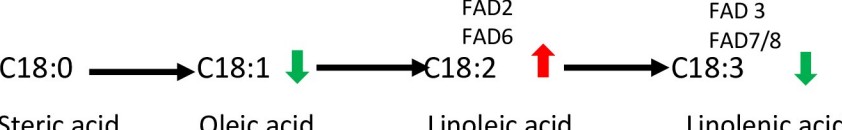

**Fig 3. Desaturation of C18 fatty acids.** This pathway is found in plastids (catalyzed by fatty acid desaturases—FAD6 and FAD7/8) and in the endoplasmic reticulum (catalyzed by FAD2 and FAD3). Arrows indicate the relative amount of each compound based on data derived from this study.

linolenic acid levels have also been identified as good markers for Brazilian *C. arabica* coffees which had lower acidity, fragrance, body and flavor [9]. Oleic acid (18:1) was 3-fold lower than linoleic acid (18:2) especially in Nia_11 (S1A Fig). Further, linolenic acid (18:3) was low (10-fold) relative to linoleic acid (18:2). It is tempting to speculate that there is differential regulation of the enzymes that catalyze these steps (Fig 3). In Arabidopsis these two fatty acid desaturases (FAD2 and FAD3) operate independently [41].

**Triacylglycerol (TG).** This group can comprise up to 75% of the lipids in coffee beans [42], and during coffee roasting, they become the carriers of the emerging flavor volatiles [4, 12]. There were 45 classes of TGs identified, varying in acyl carbons from C48 to C60 in this study. The farmer cultivated accessions i.e. the 'Niaouli' genotypes had a greater proportion of high-carbon triacylglycerols i.e. C56, C58 lipids compared to 'Kouillou' which had more C48-C52 compounds (S1B Fig). The C52 triacylglycerols showed great structural diversity containing up to six double chain triacylglycerols i.e. (52:1) to (52:6).

**Cholesterol Ester (CE).** Cholesterol esters are cholesterol molecules with a long-chain fatty acid linked to the hydroxyl group [43]. Cholesterol was thought to only be important in animal metabolism with low amounts found in plants, however, this molecule has now been identified as a key precursor for thousands of bioactive plant metabolites [44]. Cholesterol ester (20:3) was very low in abundance in *C. canephora* relative to the other lipid classes (Fig 2), however, levels were notably higher in 'Niaouli' when compared to 'Kouillou' and 'Java Robusta' (S1B Fig).

**Lysophosphatidylcholine (LPC), Phosphatidylcholine (PC) and Phosphotidylethanolamine (PE).** Phospholipids are potentially important to food sensory perception [42, 45]. PCs accumulated to higher levels than LPCs which makes sense given their relative position the lipid biosynthetic pathway (Fig 1B). Generally, the relative levels of these lipids did not vary among genotypes. PEs are usually found in plant membranes [46]. Their levels were low among the genotypes studied, but there was a significant difference found between 'Java Robusta' which did not accumulate as much PE as the other genotypes.

## Discriminatory lipid markers

**Univariate analysis.** Several studies indicate that lipids can be used as a marker to discriminate among coffee types [13, 15, 23, 47]. One-way analysis of variance ($p < 0.05$; Table 2), was used to identify lipid species that could broadly discriminate among coffees genotypes. Twenty-four lipids separated metabolites in the farmer's genotypes from those conserved at CRIN, since the data for 'Kouillou' and 'Java Robusta' were identical. 'Kouillou' and 'Java Robusta' were treated as a single group i.e. 'Kouillou/Java Robusta' because of the similarities detected. Fourteen of these lipid species were higher in 'Niaouli' and ten were higher in 'Kouillou/Java Robusta' (Table 2).

The lipid with the highest discriminatory power between the two groups of genotypes were TG (56:2), TG (58:4)A, and PE (34:2) which accumulated to high levels in 'Niaouli' relative to other genotypes. The lipid species which differentiates 'Kouillou/Java Robusta' from 'Niaouli'

**Table 2. One-way analysis of variance and Fisher's Least Square Difference (LSD) for discriminating genotypes based on lipid content.**

| Lipid species more abundant in 'Niaouli' | f.value | p.value | -LOG10(p) | FDR |
|---|---|---|---|---|
| TG (56:2) | 15.454 | 7.86E-07 | 6.1044 | 3.20E-05 |
| TG (58:4)A | 15.260 | 8.79E-07 | 6.0562 | 3.20E-05 |
| PE (34:2) | 15.037 | 1.00E-06 | 6.0000 | 3.20E-05 |
| CSH_negESI (19:2) | 11.988 | 6.84E-06 | 5.1650 | 0.000131 |
| TG (58:3) | 11.987 | 6.84E-06 | 5.1648 | 0.000131 |
| CE (20:3) | 11.212 | 1.18E-05 | 4.9299 | 0.000188 |
| CSH_posESI (11:6) | 9.9478 | 3.00E-05 | 4.5229 | 0.000411 |
| FA (28:0) | 9.0851 | 5.94E-05 | 4.2263 | 0.000713 |
| TG (58:2) | 6.4147 | 0.000645 | 3.1901 | 0.006197 |
| TG (56:3) | 5.2852 | 0.002060 | 2.6862 | 0.014124 |
| TG (56:1) | 5.1318 | 0.002431 | 2.6142 | 0.014586 |
| FA (22:0) | 5.0736 | 0.002590 | 2.5867 | 0.014627 |
| FA (20:1) | 4.4342 | 0.005305 | 2.2754 | 0.025462 |
| TG (56:4) | 3.9802 | 0.009030 | 2.0443 | 0.037692 |
| **Lipid species more abundant in 'Kouillou' /'Java Robusta'** | | | | |
| FA (16:0) | 7.3943 | 0.000255 | 3.5933 | 0.002721 |
| FA (20:0) | 6.2489 | 0.000760 | 3.1189 | 0.006637 |
| TG (50:1)_1 | 5.9483 | 0.001029 | 2.9874 | 0.008235 |
| CSH_posESI (17:6) | 5.7749 | 0.00123 | 2.9102 | 0.009082 |
| CSH_posESI (29:8) | 5.1657 | 0.002343 | 2.6302 | 0.014586 |
| CSH_posESI (04:1) | 4.6549 | 0.004124 | 2.3847 | 0.021996 |
| CSH_posESI (28:2) | 4.5120 | 0.004852 | 2.3141 | 0.024513 |
| TG (52:2) | 4.2978 | 0.006211 | 2.2069 | 0.028392 |
| TG (48:1) | 4.1161 | 0.007685 | 2.1144 | 0.033535 |
| CSH_posESI (14:1) | 3.8540 | 0.010507 | 1.9785 | 0.042028 |

The Farmer's cultivated accessions ('Niaouli' i.e. Nia_1, Nia_2, Nia_3) and those from the conserved genotypes ('Kouillou' i.e. C111, C36 and 'Java Robusta' i.e. T1049) were compared. The *f*-value is derived from the *F*. statistic test for significance, the *p*-value tests variability between two groups,–LOG$_{10}$(*p*) determines the significant levels, and FDR means False Discovery Rate.

were two saturated fatty acids, palmitic (16:0) arachidic acids (20:0) and TG (50:1). Based on our data analysis we show in Table 3, the lipids that have a high discriminatory power in the coffees in our study, were compared with information from data that was published by others. Fatty acids were the most discriminatory of all lipids, although the types varied with respect to the species examined (Table 3).

**Table 3. Fatty acids with high discriminatory power among coffee genotypes.**

| 'Niaouli' | 'Kouillou' | *C. arabica* | *C. canephora* | Specialty coffees |
|---|---|---|---|---|
| ▪ FA (22:0) Behenic acid. | ▪ FA (16:0) Palmitic acid | ▪ FA (18:2) Linoleic acid [18] | ▪ FA (18:1) Oleic acid [18] | ▪ FA (14:0) Myristic acid [6][1] |
| ▪ FA (20:1) Arachidic acid.<br>▪ FA (28:0) Montanic acid | ▪ FA (20:0) Arachidic acid | ▪ Polyunsaturated Fatty acids [16] | ▪ Monounsaturated fatty acids [18] | ▪ FA (20:0) Arachidic acid.<br>▪ FA (18:0) Stearic acid<br>▪ FA (16:0) Palmitic acid [9][2] |

1 C. canephora;

2: C. arabica.

Included are the two main varieties of *C. canephora* ('Niaouli' and 'Kouillou') used in this study, two cultivated *Coffea* species (*C. arabica* and *C. canephora*) and specialty or high-quality coffee. Citations to the referenced data are included.

**Multivariate analysis–partial least square projections.** Variable influence (or importance) of projection (VIP) scores were used as another tool to identify the lipids that contribute most to differences among genotypes (Fig 4). This analysis predicted that PE 34:2, a negative unknown lipid, CSH_negESI (19:2), and FA (28:0) i.e. dodecanoic acid could distinguish 'Niaouli' from the conserved genotypes. The ANOVA results in Table 2, also indicated that these three compounds accumulated to higher levels in the 'Niaouli' genotypes.

## Understanding the lipid networks in *C. canephora* coffee beans

We examined the lipid-to-lipid correlations among *C. canephora* beans. Many compounds in related pathways tend to occur within defined stoichiometries and kinetics parameters [48, 49]. Neighbouring compounds may be correlated with each other. This is also true of metabolites that are in chemical equilibrium, show mass conservation, or are highly sensitive to a common parameter [48, 50]. Using Pearson's correlation coefficient at a cut-off of $\geq |\pm 0.75|$, we identified those lipids that co-occurred at similar relative levels across the *C. canephora* genotypes studied. Highly specific interactions between lipids are essential for regulation of cell physiology [51, 52] (Fig 5 and detailed in S2 Table).

The most notable observation was the very strong negative correlations between CE (20:3) and other lipids, many of which were uncharacterized (Fig 5). Negative correlations were also found with CE (20:3) and TG (54:1), TG (52:1), but most notably with TG (50.1). Based on the number of carbons in their acyl chains, it seems likely that these three TGs are closely connected in the TG biosynthetic pathway. Negative correlations among metabolites are sometimes due to

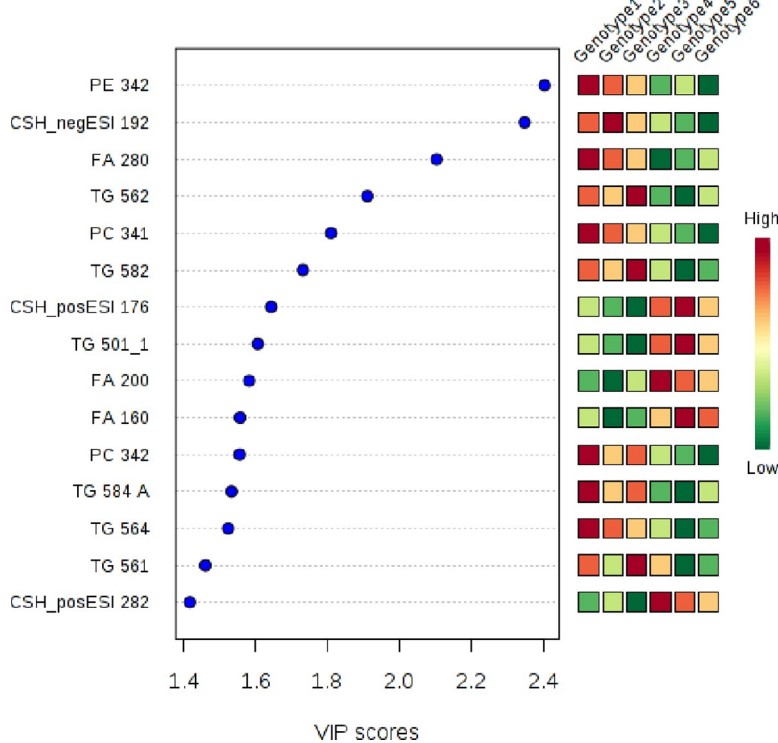

**Fig 4. Potential lipidomic markers differentiating 'Niaouli' (genotypes groups 1–3) from 'Kouillou' (groups 4–5) and 'Java' (group 6).** The variable importance in projection (VIP) scores on the x-axis provide an estimate of the contribution of a given predictor (lipid species shown on the y-axis) to the Partial Least Square (PLS) regression above. The higher the VIP score, the better the lipid species is as a predictor of the discrimination among the genotypes.

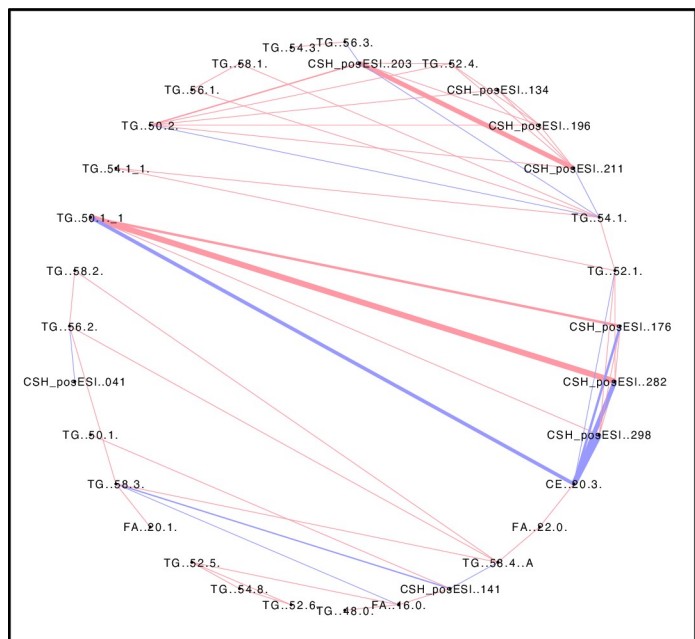

**Fig 5. Pearson's correlation network illustrating correlation patterns among lipids.** Correlations found among lipid data are shown as attribute circular layout. The blue lines indicate lipids whose occurrence negatively correlate, while the pink lines denote lipids that show a positive correlation. Line thickness indicates the strength of correlation. A permissive threshold level of $\geq |\pm 0.75|$ was set for the Pearson's correlation coefficient and a $q$FDR of $<0.05$ was chosen for the identification of significant correlations within the lipid data.

regulatory mechanisms controlling metabolites that have conserved moieties, or metabolites connected by enzymes with high variance which can cause negative correlations between its substrate and product metabolites [48, 53]. Based on the simplified pathway in Fig 1B, a relationship between the CE and TGs can be envisaged. Not all of the relationships between CE (20:3) and other lipids were negative. There was a positive correlation between CE (20:3) and TG (58:2), TG (58:4), and FA (22:0) respectively. Most triacylglycerols were positively associated with other lipids except, TG (52:0) and TG (50:1), which showed a negative correlation (Fig 5).

## Comparing the primary, secondary metabolites and lipid profiles among *C. canephora*

Metabolomics and lipidomics are powerful strategies to qualitatively and quantitatively analyze a wide range of small molecules in a biological sample, which represent endpoints of genome expression [54]. The metabolomics data from our recent work [24] was subjected to PLS-DA analysis for both primary and secondary metabolites and compared to the lipidomics profile of these coffees. Our aim was to determine the diversity of metabolism across genotypes. We expected greater similarity in primary metabolism across genotypes, and that specialized metabolites would be more divergent [55]. Lipids are primary metabolites but in coffee beans, it is not known the extent to which they are part of the fundamental or specialized metabolic machinery.

We found that there was clearer differentiation between 'Niaouli' genotypes which clustered together and away from the 'Kouillou' and 'Java Robusta' when secondary metabolites were examined (Fig 6B) compared to primary metabolites (Fig 6A). No clear difference was observed among the groups with PLS-DA on lipid (Fig 6C) but SPLS-DA—(Fig 6D) was able to reveal more similarity between 'Java' and the 'Niaouli' types, while the 'Kouillou' genotypes were separate from the rest. This is noteworthy because of the many studies that show 1) that

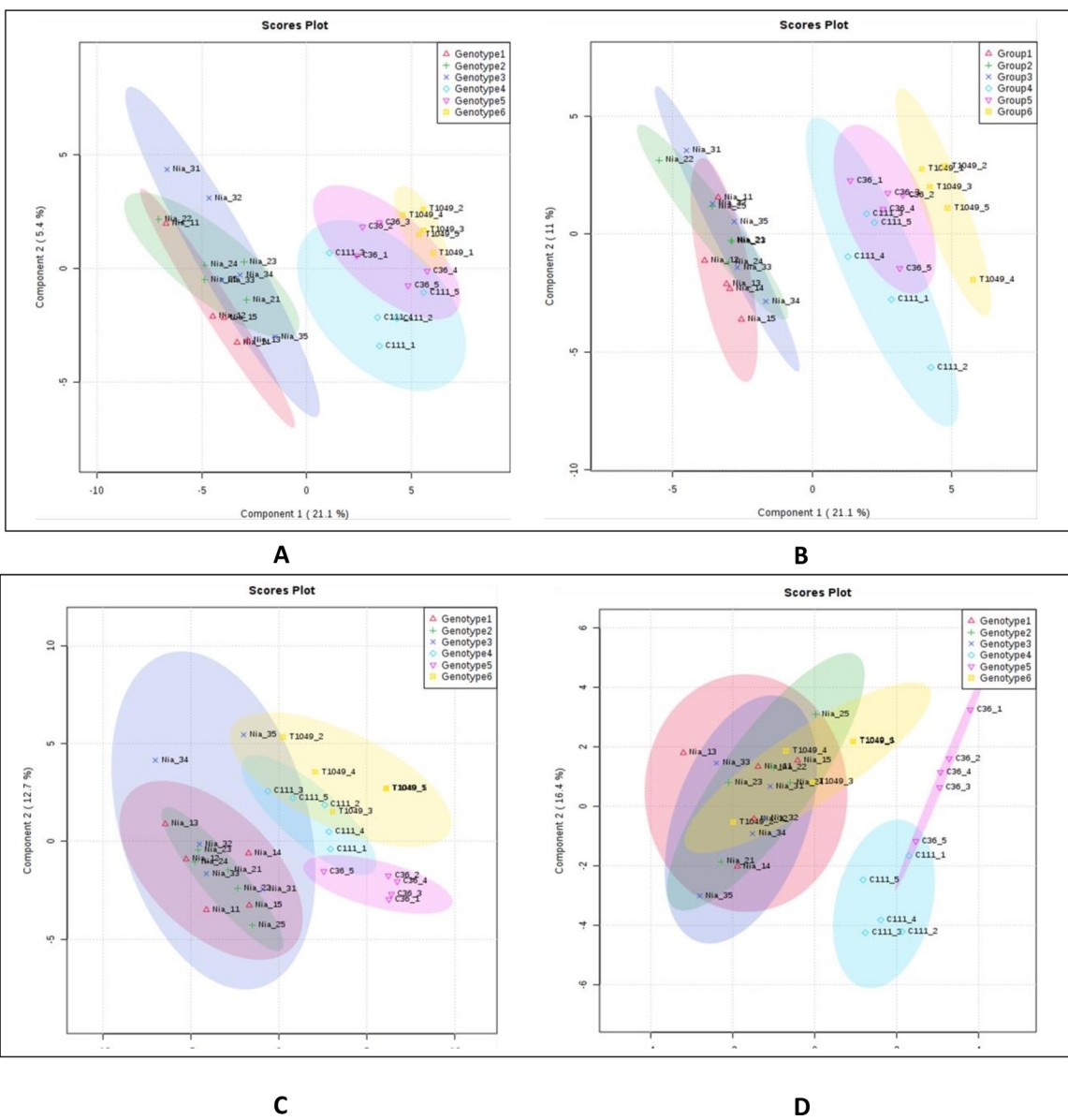

**Fig 6. Differentiating among the *C. canephora* genotypes using multivariate analyses.** A) Primary metabolites analyzed by PLS-DA, B) Secondary metabolites analyzed by PLS-DA, C) lipids analyzed by PLS and D) lipids analyzed by Sparse-PLS. (PCA = Principal Component Analysis, PLS-DA = Partial Least Square–discriminate analysis. Genotypes 1–3 belong to the 'Niaouli' group i.e. Nia_1, Nia_2 and Nia_3, Genotypes 4–5 are members of the 'Kouillou' group i.e. C111 and C36, and Genotype 6 is 'Java Robusta' i.e. T1049. The primary and secondary metabolite data were derived from Anagbogu *et al*., 2019b [24].

lipids are valuable components of the sensory attributes of coffee and 2) that they are excellent at discriminating genotypes and growing environment [8, 9, 14].

## Correlating lipidomic profiles to genotypic variation of coffee

In spite of the importance of lipids in coffee cup quality improvement, as far as we know, it has not been fully integrated into coffee breeding strategies except in the study by Sant'Ana et al. [23]. The extracted DNA of the coffee genotypes in this study was analyzed using Genotype-by-Sequencing and their relatedness compared using hierarchical clustering. The lipidomics

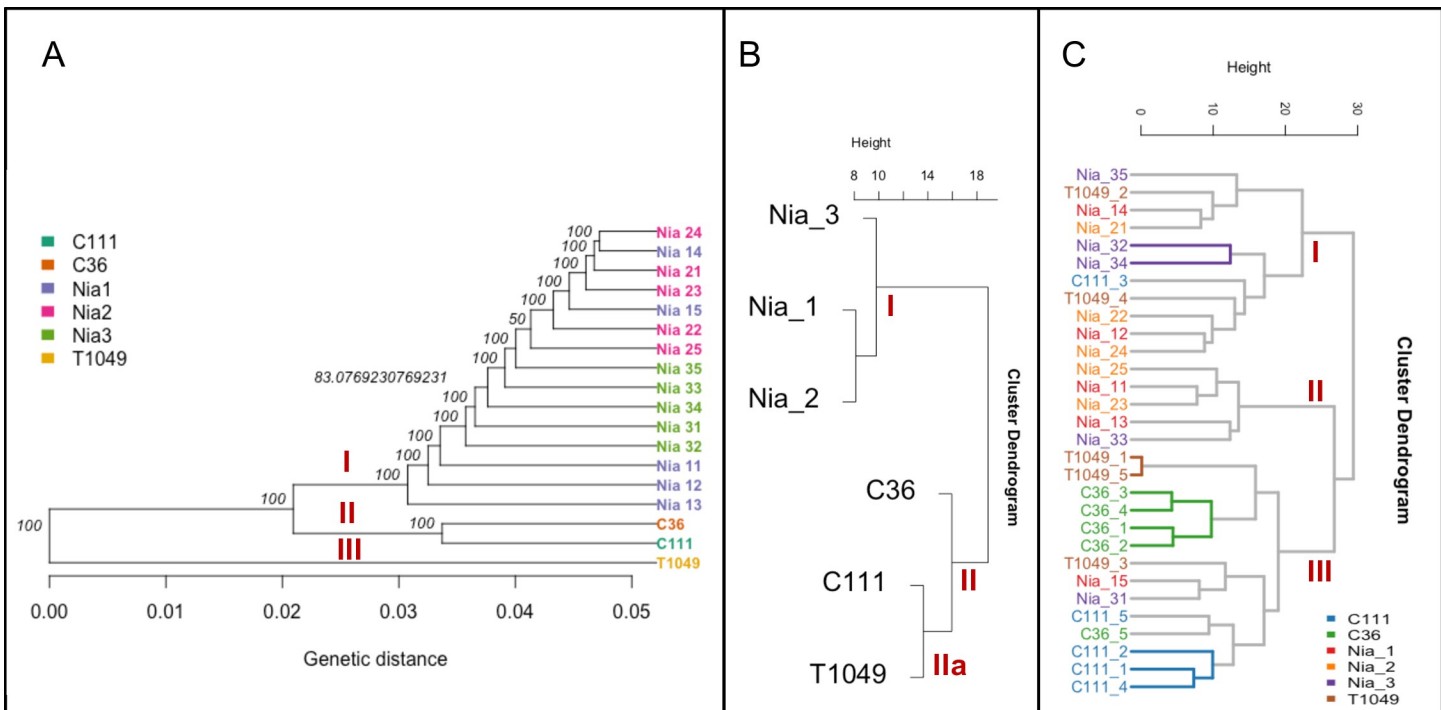

**Fig 7. Comparative analysis of the genomics and lipidomic analysis based on their hierarchical structure.** A) genetic analysis based on ~100,000 SNP, B) lipidomic analysis indicating the mean value of each genotype from five biological replicates and C) lipidomic analysis including all replicates. The genotypes and replicates names were described in Table 1 as follows: Nia_11- Nia_15 were Genotype 1, Nia_21-Nia_25 were genotype 2, Nia_31-Nia35 were Genotype 3 and belong to the 'Niaouli' group; Members of the 'Kouillou' group have been referred to as Genotypes 4 and 5 in this study and included C111_1-C111_5 and C36_1 to C36_5. 'Java Robusta' coffee has been described as Genotype 6 included T1049_1-T1049_5.

data was also analyzed similarly. Overall, there was good agreement between the DNA- and metabolite-based clustering (Fig 7A). From the genetic perspective, we observed three clusters I, II and III representing 'Niaouli', 'Kouillou', and 'Java' respectively (Fig 7A). The lipidomic analysis of the genotypes based on the mean values (Fig 7B) revealed two main clusters, separating the 'Niaouli' from 'Kouillou'/'Java Robusta' respectively (Fig 7B). However there were notable exceptions: SNP analysis indicated that C36 and C111 ('Kouillou') were highly related, while T1049 ('Java Robusta') was divergent, in contrast, lipidomics indicted that T1049 and C111 formed a distinguishable subgroup (cluster IIa) while C36 was more distant (Fig 7C).

All lipid analyses (Fig 7B and 7C) revealed a clear disparity between the 'Niaouli' and 'Kouilou/ Java Robusta' genotypes. We suggest that there are two lipid diversity structures within the population studied which can be used for future coffee improvement in Nigeria. From this study, FA(16:0) and FA(20:0) were higher ($p<0.05$) in the 'Kouillou' genotypes (Table 2), and these fatty acids are among the precursors of good quality coffee (Table 2). A previous study identified the fatty acid desaturase (*FAD2*) encoded gene as one lipid-associated SNP in coffee [23]. Further study of our SNP data to detect the sources or causes of the distinct lipid difference between 'Kouillou' and 'Niaouli' is needed. For example, incorporating alleles for higher accumulation of FA(16:0), which is associated with high sensory attributes into the genome of cultivated variety ('Niaouli') (Table 2) would be of great help towards its quality improvement.

## Conclusion

There is much interest in using lipid profiles to identify biomarkers for coffee cup quality improvement. In this study, we used various data mining tools to better understand variations

in lipid profiles of Southwestern Nigerian coffee genotypes, and how these lipids relate to other metabolites, and to other studies done on coffee. Insight into the gene expression of coffee bean lipids would help to elucidate if there was an environmental effect on relative lipid abundance and in the proper identification of candidate genes. Despite this limitation, our findings can be summarized as following: 1) Lipids that could discriminate among the genotypes studied were identified, even though there was less variability for lipids among genotypes compared with other types of metabolites. 2) 'Java Robusta' and 'Kouillou' especially 'Kouillou' genotype C36, contained more saturated fatty acyls species such as FA(26:0), FA(16:0), FA (20:0), while 'Niaouli' contained more unsaturated FAs such as FA(16:1), FA(20:1). It should be interesting to study the SNPs of gene such as FADs which are responsible for desaturation in coffee bean. 3) Unexpected observations were made with respect to cholesterol ester (20:3) which was one of the least abundant lipids in our analysis. The levels of CE(20:3) and some TG compounds, showed some of the strongest negative correlations. Further, CE(20:3) can discriminate between 'Niaouli' and ''Kouillou' and Java Robusta' indicating its value as a genotypic marker. 4) Lipids as a class of compounds in this analysis did not discriminate among the genotype studied, as well as secondary metabolites but grouped genotypes into two lipid profile structures. 5) There was a good correlation between the classification of genotypes based on SNP variability, and the lipidomic profile. In addition, the two divergent lipid profiles correlating with the *C. canephora* genotypes will be tapped and included in coffee cup quality improvement program. The lipid species TG (56:2), TG (58:4), and PE (34:2) showed the highest discriminatory power in distinguishing 'Niaouli' from 'Kouillou/Java Robusta'. However, the saturated fatty acids, palmitic (16:0) and arachidic acids (20:0), were higher in 'Kouillou/Java Robusta'. Hybridization between the two groups of coffee would be highly recommended to generate genotypes with a more favorable lipid profile for coffee cup quality improvement. This study was an important starting point for coffee quality improvement in Nigeria. Although we found some divergent lipids, generally, lipid diversity was low among the genotypes studied, and there will be a need to initiate genetic acquisition from other coffee producing countries through collaboration.

## Supporting information

**S1 Fig. The relative amounts of the different groups of lipids in the coffee.**
(PDF)

**S1 Table. The lipid species identified with corresponding charge mode.**
(PDF)

**S2 Table. Pearson's correlation coefficient values, P-values and adjusted P-values for lipid-to-lipid associations.**
(PDF)

## Acknowledgments

We appreciate the International Institute of Tropical Agriculture and the University of California Davis, USA, for providing free access to their facilities. We also thank Drs. May Thitisaksakul, Karin Albornoz, Shaoyun Dong, and Jorge Dubcovsky at UC Davis, USA, for their advice and technical support.

## Author Contributions

**Conceptualization:** Chinyere F. Anagbogu, Diane M. Beckles.

**Formal analysis:** Chinyere F. Anagbogu, Jiaqi Zhou, Diane M. Beckles.

**Funding acquisition:** Chinyere F. Anagbogu, Diane M. Beckles.

**Investigation:** Chinyere F. Anagbogu, Diane M. Beckles.

**Methodology:** Chinyere F. Anagbogu, Festus O. Olasupo, Mohammed Baba Nitsa, Diane M. Beckles.

**Project administration:** Diane M. Beckles.

**Supervision:** Diane M. Beckles.

**Validation:** Jiaqi Zhou.

**Visualization:** Jiaqi Zhou, Diane M. Beckles.

**Writing – original draft:** Chinyere F. Anagbogu.

**Writing – review & editing:** Jiaqi Zhou, Festus O. Olasupo, Mohammed Baba Nitsa, Diane M. Beckles.

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
