## [Decision Letter · Decision Letter 0]

20 Aug 2020

PONE-D-20-16481

Lipidomic and metabolomic profiles of Coffea canephora L. beans cultivated in South-western Nigeria

PLOS ONE

Dear Dr. Florence,   

Thank you for submitting your manuscript to PLOS ONE. After careful consideration, we feel that it has merit but does not fully meet PLOS ONE’s publication criteria as it currently stands. Therefore, we invite you to submit a revised version of the manuscript that addresses the points raised during the review process.

We look forward to receiving your revised manuscript.

Kind regards,

Ch Ratnasekhar, Ph.D.

Academic Editor

PLOS ONE

Journal Requirements:

'Support for this research was provided in part by the Norman E. Borlaug Leadership Enhancement in Agriculture Program (Borlaug LEAP) through a grant to the University of California-Davis by the United States Agency for International Development. The opinions expressed herein are those of the authors and do not necessarily reflect the views of USAID. Funding was also provided by the Federal Ministry of Agriculture in Nigeria through a grant to the Cocoa Research Institute of Nigeria.'

'The funders had no role in study design, data collection and analysis, decision to publish, or preparation of the manuscript.'

Please clarify the sources of funding (financial or material support) for your study. List the grants or organizations that supported your study, including funding received from your institution.State what role the funders took in the study. If the funders had no role in your study, please state: “The funders had no role in study design, data collection and analysis, decision to publish, or preparation of the manuscript.”If any authors received a salary from any of your funders, please state which authors and which funders.

4. Please amend the manuscript submission data (via Edit Submission) to include author Chinyere F. Anagbogu and amend your authorship list in your manuscript file to include author Florence F. Anagbogu.

Reviewers' comments:

Reviewer's Responses to Questions

**Comments to the Author**

1. Is the manuscript technically sound, and do the data support the conclusions?

Reviewer #1: Yes

Reviewer #2: Yes

2. Has the statistical analysis been performed appropriately and rigorously? 

Reviewer #1: Yes

Reviewer #2: Yes

3. Have the authors made all data underlying the findings in their manuscript fully available?

Reviewer #1: No

Reviewer #2: Yes

4. Is the manuscript presented in an intelligible fashion and written in standard English?

Reviewer #1: Yes

Reviewer #2: Yes

5. Review Comments to the Author

Reviewer #1: The manuscript entitled “Lipidomic and metabolomic profiles of Coffea canephora L. beans cultivated in South-western Nigeria” has been written and presented well. The work in the manuscript describes the untargeted lipid profiling of coffee seeds collected from South-western Nigeria using high resolution mass spectrometry. The lipidomic and metabolic profiling have been compared and correlated with the genotypic variation of the coffee from three different regions. The statistical analysis and results presented in the manuscript is appreciable.

Some of my specific comments/queries to improve the manuscript are as follows:

Comment 1.

In Introduction section, please add a paragraph about correlation between metabolic profiling and genotypic study.

Comment 2.

Line 123-124, Why internal standard was use at the time of reconstitution? And not during extraction procedure?

Comment 3.

Line 200-202, Use same numerical representation for the number of lipids identified in different classes.

Comment 4.

Author should add in abstract and conclusion section about scientific novelty of the work presented in the manuscript.

Comment 5.

Write a separate paragraph about the earlier reported work in the field of lipidomic profiling and their correlation with genotypic variation of coffee seeds along with limitation of the present study.

Comment 6.

Please provide some chromatograms obtained from LC-MS/MS analysis in the supplementary information.

Comment 7.

Please provide sequencing data of all the coffee variants compared and provide some discussion about it.

Comment 7.

How this study can improve the quality of the coffee?

Reviewer #2: The manuscript is well written but my only concerns are that: since Nigeria is not a major coffee producing country it could have been excellent if coffee from other major coffee producing countries (one) is used as reference?

Abstract

Line 35 could be ''popular refreshing beverage globally.

Materials

Line 133 how many samples were collected from farmers & how many from CRIN, Was there any evidence of difference due to locations?:

Conclusion

line 397 kindly specify the statement lipids that could discriminate among the genotypes

6. PLOS authors have the option to publish the peer review history of their article (what does this mean?). If published, this will include your full peer review and any attached files.

Reviewer #1: No

Reviewer #2: No

---

## [Author Response · Author response to Decision Letter 0]

9 Nov 2020

Comments to the Author

1. Is the manuscript technically sound, and do the data support the conclusions?

Reviewer #1: Yes

Reviewer #2: Yes

Response 1: Thank you

2. Has the statistical analysis been performed appropriately and rigorously?

Reviewer #1: Yes

Reviewer #2: Yes

Response 2: Thank you

3. Have the authors made all data underlying the findings in their manuscript fully available?

Reviewer #1: No

Reviewer #2: Yes

Response 3: The data has been successfully uploaded to the Dryad repository (https://doi.org/10.5061/dryad.j6q573nc4). And this has been inserted in lines 169-170. 

“The raw data from the UHPLC has been submitted to Dryad repository and can be found at the following location: https://doi.org/10.5061/dryad.j6q573nc4”

4. Is the manuscript presented in an intelligible fashion and written in standard English?

Reviewer #1: Yes

Reviewer #2: Yes

Response 4: Thank you

5. Review Comments to the Author

Response to Reviewer 1 Comments

Reviewer #1: The manuscript entitled “Lipidomic and metabolomic profiles of Coffea canephora L. beans cultivated in South-western Nigeria” has been written and presented well. The work in the manuscript describes the untargeted lipid profiling of coffee seeds collected from South-western Nigeria using high resolution mass spectrometry. The lipidomic and metabolic profiling have been compared and correlated with the genotypic variation of the coffee from three different regions. The statistical analysis and results presented in the manuscript is appreciable.

Some of my specific comments/queries to improve the manuscript are as follows:

Comment 1.

In Introduction section, please add a paragraph about correlation between metabolic profiling and genotypic study.

Response 1: Thank you, paragraph has been inserted in lines 76-83.

Comment 2.

Line 123-124, Why internal standard was use at the time of reconstitution? And not during extraction procedure?

Response 2: Thank you for catching this error, you are correct, internal standards were used during extraction, this has been changed in lines 131-133 and these references listed below included: 

1. Cajka, Tomas, Jennifer T. Smilowitz, and Oliver Fiehn, “Validating quantitative untargeted Lipidomics across nine liquid chromatography-high-resolution mass spectrometry platforms.” Analytical chemistry 89.22 (2017): 12360-12368.

2. Cajka, Tomas and Oliver Fiehn. Increasing lipidomics coverage by selecting optimal mobile-phase modifiers in LC-MS of blood plasma. Metabolomics. 2016. 12(2):34

Comment 3.

Line 200-202, Use same numerical representation for the number of lipids identified in different classes.

Response 3: Thank you. We use the APA academic writing style which states that “words should be used for numbers from zero through nine, and numerals should be used from 10 onwards.”

Comment 4.

Author should add in abstract and conclusion section about scientific novelty of the work presented in the manuscript.

Response 4: Thank you, this has been included in lines 50-52. 

“The divergent lipid profiles identified among the C. canephora genotypes, correlated with their Single Nucleotide Polymorphism grouping as assessed by genotype-by-sequencing, and will be exploited to improve coffee cup quality”

and in lines 422-432 “There was a good correlation between the classification of genotypes based on SNP variability, and the lipidomic profile. In addition, the two divergent lipid profiles correlating with the C. canephora genotypes will be tapped and included in coffee cup quality improvement program. The lipid species TG (56:2), TG (58:4), and PE (34:2) showed the highest discriminatory power in distinguishing ‘Niaouli’ from ‘Kouillou/Java Robusta’. However, the saturated fatty acids, palmitic (16:0) and arachidic acids (20:0), were higher in ‘Kouillou/Java Robusta’. Hybridization between the two groups of coffee would be highly recommended to generate genotypes with a more favorable lipid profile for coffee cup quality improvement. This study was an important starting point for coffee quality improvement in Nigeria. Although we found some divergent lipids, generally, lipid diversity was low among the genotypes studied, and there will be a need to initiate genetic acquisition from other coffee producing countries through collaboration”.

Comment 5.

Write a separate paragraph about the earlier reported work in the field of lipidomic profiling and their correlation with genotypic variation of coffee seeds along with limitation of the present study.

Response 5: Thank you, this has been included in the result and discussion section under, “Correlating lipidomic profiles to genotypic variation of coffee” in line 373.

Comment 6.

Please provide some chromatograms obtained from LC-MS/MS analysis in the supplementary information.

Response 6: The lipidomics work was performed at the NIH West Coast Metabolomic Center at the University of California Davis. We tried to obtain chromatograms from the analysis but it was not successful. The following references contain details of the method used to generate our data.

Cajka, Tomas, Jennifer T. Smilowitz, and Oliver Fiehn, “Validating quantitative untargeted Lipidomics across nine liquid chromatography-high-resolution mass spectrometry platforms.” Analytical chemistry 89.22 (2017): 12360-12368.

Cajka, Tomas and Oliver Fiehn. Increasing lipidomics coverage by selecting optimal mobile-phase modifiers in LC-MS of blood plasma. Metabolomics. 2016. 12(2):3

Comment 7.

Please provide sequencing data of all the coffee variants compared and provide some discussion about it.

Response 7: Thank you, the DNA sequencing data can be found in our recent publication:

 Anagbogu CF, Bhattacharjee R, Ilori C, Tongyoo P, Dada KE, Muyiwa AA, et al. Genetic diversity and re-classification of coffee (Coffea canephora Pierre ex A. Froehner) from South Western Nigeria through genotyping-by-sequencing-single nucleotide polymorphism analysis. Genet Resour Crop Ev. 2019;66(3):685-96. PubMed PMID: WOS:000461347800009.

It has also been deposited at National Center for Biotechnology Information (NCBI), Sequence Read Archive (study accession number SRP096172: https://www.ncbi.nlm.nih.gov/sra/?term=SRP096172)

Comment 8.

How this study can improve the quality of the coffee?

Response 8: Thank you, this has been elaborated and inserted in lines 422-432 

“There was a good correlation between the classification of genotypes based on SNP variability, and the lipidomic profile. In addition, the two divergent lipid profiles correlating with the C. canephora genotypes will be tapped and included in coffee cup quality improvement program. The lipid species TG (56:2), TG (58:4), and PE (34:2) showed the highest discriminatory power in distinguishing ‘Niaouli’ from ‘Kouillou/Java Robusta’. However, the saturated fatty acids, palmitic (16:0) and arachidic acids (20:0), were higher in ‘Kouillou/Java Robusta’. Hybridization between the two groups of coffee would be highly recommended to generate genotypes with a more favorable lipid profile for coffee cup quality improvement. This study was an important starting point for coffee quality improvement in Nigeria. Although we found some divergent lipids, generally, lipid diversity was low among the genotypes studied, and there will be a need to initiate genetic acquisition from other coffee producing countries through collaboration”.

Response to Reviewer #2 Comments

Comment 1: The manuscript is well written but my only concerns are that: since Nigeria is not a major coffee producing country it could have been excellent if coffee from other major coffee producing countries (one) is used as reference?

Response 1: Thank you; the work in this study aims to establish Nigeria as an important coffee-producing region. The main coffee species cultivated in the country is C. canephora, which is more resistant to abiotic and biotic stress compared to C.arabica. Because generally, C. canephora differs in quality to the well-established C.arabica, it is necessary to begin the basic characterization necessary for examining cup quality in these coffees and to determine how our Nigerian genotypes compare to those studied. Throughout the manuscript we have compared our results to those generated in other more popular coffee-producing countries. Examples of work referenced in our manuscript are given below. 

Speer K, Kölling-Speer I. The lipid fraction of the coffee bean. Brazilian Journal of Plant Physiology. 2006;18:201-16.

Figueiredo LP, Borém FM, Ribeiro FC, Giomo GS, Henrique da Silva Taveira J, Malta MR. Fatty acid profiles and parameters of quality of specialty coffees produced in different Brazilian regions. African Journal of Agricultural Research. 2015;10(35):3484-93. doi: https://doi.org/10.5897/AJAR2015.9697.

Navarini L, Ferrari M, Liverani FS, Liggieri L, Ravera F. Dynamic tensiometric characterization of espresso coffee beverage. Food Hydrocolloid. 2004;18(3):387-93. PubMed PMID: WOS:000220352600005.

Mehari B, Redi-Abshiro M, Chandravanshi BS, Combrinck S, McCrindle R, Atlabachew M. GC-MS profiling of fatty acids in green coffee (Coffea arabica L.) beans and chemometric modeling for tracing geographical origins from Ethiopia. Journal of the Science of Food and Agriculture. 2019;99(8):3811-23. doi: 10.1002/jsfa.9603.

Please also look at Table 4, where we compared fatty acids from our study with those from two C. arabica, C. canephora, and specialty or high-quality coffee, from other studies along with citations. Still, there have not been extensive lipidomics data published in C. canephora limiting the extent to which those comparisons could be made. This perhaps highlights the novelty of our work.

Comment 2: Abstract

Line 35 could be ''popular refreshing beverage globally.

Response 2: Thank you for this suggestion, the text has been changed in line 35

Comment 3: Materials

Line 133 how many samples were collected from farmers & how many from CRIN? 

Response 3: Thank you, this information on sampling has been included in lines 114-117. See below”

“Niaouli’ is comprised of three genotypes collected from different farmers’ field: Nia_1, Nia_2 and Nia_3, (respectively, classified as Groups 1, 2 and 3). ‘Kouillou’ is comprised of two genotypes from CRIN: C111 and C36, (respectively, classified as Groups 4, and 5) and, ‘Java Robusta’ is comprised of one genotype from CRIN: T1049, and was classified as Group 6. There were five replicates for each group (genotype) giving a total of 30 samples (Table 1).”

Was there any evidence of difference due to locations?

It seems likely that the genotypes we studied were adapted to different regions. There were altitudinal differences between the farmers’ field (433-522 m above sea level) which consisted of one variety ‘Niaouli’, and the CRIN plantation, (133 m above sea level) where ‘Kouilou’ and ‘Java’ are cultivated. The farmers’ genotypes ‘Niaouli’ were genetically distinct from the two other varieties conserved in CRIN, based on our previous SNP-GBS analysis [23]. The lipidomics data largely followed this classification.

[23] Anagbogu CF, Bhattacharjee R, Ilori C, Tongyoo P, Dada KE, Muyiwa AA, et al. Genetic diversity and re-classification of coffee (Coffea canephora Pierre ex A. Froehner) from South Western Nigeria through genotyping-by-sequencing-single nucleotide polymorphism analysis. Genet Resour Crop Ev. 2019;66(3):685-96. PubMed PMID: WOS:000461347800009.

This statement below was also included and inserted in lines 118-121.

“Both farmers’ fields and CRIN were located in south-western Nigeria [23], with altitude of 433-522 m and 133 m above sea level were recorded using ArcGIS software (Redlands, California) respectively. There longitude and latitude coordinates could also be found in Anagbogu et al. 2019”.

Comment 4: Conclusion

line 397 kindly specify the statement lipids that could discriminate among the genotypes

Response 4: Thank you, this has been inserted in lines 425-427 

“The lipid species, TG (56:2), TG (58:4), and PE (34:2) showed the highest discriminatory power in distinguishing ‘Niaouli’ from ‘Kouillou/Java Robusta’. However, the saturated fatty acids, palmitic (16:0) and arachidic acids (20:0), were higher in ‘Kouillou/Java Robusta’.

---

## [Decision Letter · Decision Letter 1]

16 Dec 2020

Lipidomic and metabolomic profiles of Coffea canephora L. beans cultivated in South-western Nigeria

PONE-D-20-16481R1

Dear Dr. Florence,

We’re pleased to inform you that your manuscript has been judged scientifically suitable for publication and will be formally accepted for publication once it meets all outstanding technical requirements. 

Kind regards,

Ch Ratnasekhar, Ph.D.

Academic Editor

PLOS ONE

Additional Editor Comments (optional):

Please provide the LC-MS/MS chromatograms of lipids in the supplementary. 

Reviewers' comments:

Reviewer's Responses to Questions

**Comments to the Author**

1. If the authors have adequately addressed your comments raised in a previous round of review and you feel that this manuscript is now acceptable for publication, you may indicate that here to bypass the “Comments to the Author” section, enter your conflict of interest statement in the “Confidential to Editor” section, and submit your "Accept" recommendation.

Reviewer #1: All comments have been addressed

Reviewer #2: All comments have been addressed

2. Is the manuscript technically sound, and do the data support the conclusions?

Reviewer #1: Yes

Reviewer #2: Yes

3. Has the statistical analysis been performed appropriately and rigorously? 

Reviewer #1: Yes

Reviewer #2: Yes

4. Have the authors made all data underlying the findings in their manuscript fully available?

Reviewer #1: No

Reviewer #2: No

5. Is the manuscript presented in an intelligible fashion and written in standard English?

Reviewer #1: Yes

Reviewer #2: Yes

6. Review Comments to the Author

Reviewer #1: All the comments and queries raised have been addressed by the authors. However, authors have not been able to provide chromatograms of lipids obtained from LC-MS/MS which is still questionable. I would like to recommend the proposed work to be published in PLOS ONE, if they can provide chromatograms obtained of LC-MS/MS as supplementary file.

Reviewer #2: All my comments have been fully addressed. Good work done to improve and put Nigeria coffee on the globe

7. PLOS authors have the option to publish the peer review history of their article (what does this mean?). If published, this will include your full peer review and any attached files.

Reviewer #1: No

Reviewer #2: **Yes: **Ernest Teye

---

## [Editor Report · Acceptance letter]

5 Feb 2021

PONE-D-20-16481R1 

Lipidomic and metabolomic profiles of *Coffea canephora* L. beans cultivated in South-western Nigeria 

Dear Dr. Anagbogu:

I'm pleased to inform you that your manuscript has been deemed suitable for publication in PLOS ONE. Congratulations! Your manuscript is now with our production department. 

Kind regards, 

on behalf of

Dr. Ch Ratnasekhar 

Academic Editor

PLOS ONE